

# Probability estimation of March 1989-like geomagnetic storms and their relevance for the insurance industry

Deniz Güney Akkor[1], Prof. Dr. Halit Ünal Özden [2]

[1]Insurance and Risk Management, Istanbul Commerce University, Munich, 81829, Germany
[2]Istanbul Commerce University, Istanbul, 34445, Türkiye

*Correspondence to*: Deniz Güney Akkor (d.g.akkor@gmail.com)

**Abstract.** This study employs Extreme Value Theory (EVT) to estimate the probability of geomagnetic storms of comparable magnitude to the March 1989 event and to assess the implications of such storms for the insurance industry. To calculate return periods for extreme events, historical Dst (Disturbance Storm Time) data from the World Data Centre for Geomagnetism are

combined with the Generalized Extreme Value (GEV) distribution, maximum likelihood estimation, and the Peaks Over Threshold (POT) approach. The findings suggest that there is a 7.14% to 8.33% chance of a geomagnetic storm of equivalent severity occurring during the next 70 years (with a 95% confidence interval). This study helps us understand the frequency and severity of extreme geomagnetic storms and helps the insurance industry make judgments about risk management.

## 1 Introduction

### 1.1 Background and motivation

Geomagnetic storms, such as the March 1989 event, have demonstrated significant potential to disrupt modern technology and infrastructure. The resulting economic impacts have prompted interest within the insurance industry, calling for better risk assessment and management techniques (Brynjolfsson and Hitt, 2000). Geomagnetic storms are alterations in the magnetosphere of the Earth that can seriously harm technological infrastructure and interfere with communication systems.

Therefore, in order to minimize potential harm and safeguard crucial infrastructure, it is crucial to conduct precise and timely evaluations of geomagnetic storm risk. Several methods can be used to achieve this, such as modelling the anticipated effects of prospective space weather events on the Earth's magnetic field and keeping an eye on solar and planetary conditions to spot them. The Kp and Dst indices can also be used to measure changes in the Earth's magnetic field's horizontal component and gauge the geomagnetic storm's intensity. A geomagnetic storm risk assessment must also take into consideration risk elements

like geomagnetic latitude, ground conductivity, coastal effects, and transmission system characteristics.

Additionally, the analysis of the power grid model gives risk managers a crucial tool for evaluating system-level effects and locating vulnerable locations. To minimize potential harm and safeguard crucial infrastructure, it is essential to adopt a




thorough and multifaceted approach to geomagnetic storm risk assessment. In general, it is crucial to be aware of the catastrophic repercussions that geomagnetic storms may have and to take proactive steps to address potential dangers through
thorough risk assessment and mitigation techniques.

The difficulty of controlling risks brought on by prospective extreme weather occurrences, such as geomagnetic storms, is one that the insurance business must overcome. Power grids, communication networks, and other crucial infrastructure can all be severely disrupted by geomagnetic storms. Furthermore, these storms have the potential to produce geomagnetically induced currents, which could harm transformers and other electrical machinery. Such damage entails enormous financial risks for
insurance providers and their customers, which include organizations, governments, and people. Furthermore, it is critical to precisely evaluate the likelihood of geomagnetic storms due to the potential effects on economic activity and societal well-being.

This study seeks to address this issue by estimating the chance and return period of geomagnetic storms of a size comparable to the event of March 1989 and evaluating their potential effects on the insurance sector.

**1.2 Objectives of the study**

The main goal of this work is to use Extreme Value Theory (EVT) to calculate the likelihood of geomagnetic storms of a similar intensity to the event in March 1989. Through an analysis of the implications of our findings for risk assessment and management, we also seek to determine the relevance of these storms for the insurance sector. In order to provide a more thorough risk assessment, the study also aims to emphasize the significance of implementing a holistic strategy that considers
a variety of elements, such as power grid models, geological influences, and transmission system features.

**1.3 Structure of the paper**

The article is set up like follows a review of the literature is presented in Section 2 and covers Weibull law, historical Dst data, EVT applications in natural catastrophes, geomagnetic storms and their impacts, and risk assessment in the insurance sector. The methodology is presented in Section 3, which also covers the data sources and preprocessing, fitting the GEV distribution,
maximum likelihood parameter estimation, and return level analysis. Results are presented in Section 4 together with fitted GEV distribution parameters, predicted return times, a probabilistic evaluation of a geomagnetic storm comparable to March 1989, sensitivity analysis, and validation. The results are discussed, compared to earlier studies, and their implications for risk management are assessed in Section 5. The conclusion concludes by summarizing the key findings and their significance for the insurance sector.



## 2 Literature Review

### 2.1. Geomagnetic storms and their effects

Geomagnetic storms are alterations in the magnetic field of the Earth brought on by solar activity such as solar flares and coronal mass ejections. Alternatively said, they are a significant disturbance in the Earth's magnetosphere that happens when there is a strong energy exchange from the solar wind into the space environment around the Earth. Variations in the solar wind that result in significant changes in the currents, plasmas, and fields in the magnetosphere of the Earth are what lead to these storms. Geomagnetic storms can cause significant disruptions to power grids, satellite systems, and communication networks (Boteler, 2001; Pulkkinen et al., 2017). Numerous researches have looked at how geomagnetic storms affect different businesses and infrastructure. For instance, Boteler examined the impact of geomagnetic storms on electrical networks and discovered that, under rare circumstances, widespread blackouts can happen. The threats that geomagnetic storms represent to numerous space-based assets, including satellites and navigation systems, were also investigated by Pulkkinen et al.

### 2.2. Historical Dst data

A global geomagnetic indication called Dst (Disturbance Storm Time) assesses the intensity of geomagnetic storms. The World Data Center for Geomagnetism in Kyoto offers historical Dst data, which are crucial for determining the frequency and power of previous geomagnetic storms.

Using sources like Boteler and Pulkkinen, the literature review for this paper gives an overview of geomagnetic storms and their impacts. The analysis emphasizes the enormous interference that geomagnetic storms have on electrical grids, and communication networks. It also highlights the value of historical Dst data in determining the frequency and strength of previous geomagnetic storms. This assessment also demonstrates the importance of researching geomagnetic storms and their effects on the insurance sector.

### 2.3. Extreme Value Theory (EVT) applications in natural hazards

EVT is a statistical framework for simulating the frequency and size of rare and severe events. EVT has become instrumental in assessing the risk of extreme natural events, allowing for better predictability and preparedness (Coles, 2001; Katz et al., 2002).

Extreme Value Theory is used in this study to model the frequency and size of extreme geomagnetic storms. Coles and Katz et al. have already shown how to use EVT for the analysis of natural hazards, including geomagnetic storms. This study intends to advance knowledge of the likelihood and potential effects of extreme geomagnetic storms on numerous infrastructure and industry sectors through its application. As demonstrated in the works of Boteler and Pulkkinen, the literature evaluation for this study underlines the major disruptions caused by geomagnetic storms to power grids, communication systems, and satellite



operations. Additionally, for the purpose of managing insurance risk, this work aims to apply EVT to geomagnetic storms
while acknowledging its value in simulating extreme natural hazards.

## 2.4. Weibull Law

A flexible probability distribution used to simulate different types of data is the Weibull distribution. As detailed by Weibull, the statistical distribution function offers extensive applicability, which has made it an essential reference in studying extreme events (Weibull, 1951). The Weibull Law, a flexible probability distribution used to describe different forms of data, has been 90   frequently utilized in the context of extreme value theory to estimate the likelihood that extreme occurrences, such geomagnetic storms, will exceed predetermined thresholds. The Weibull Law has the capacity to accurately predict extreme geomagnetic storms and the risks they pose. The goal of this work is to better understand the possible effects of extreme geomagnetic storms on various businesses and infrastructure, particularly the insurance sector, by modeling the exceedance probability of such storms using the Weibull Law. In earlier research, such as those by Dahen et al. and Rootzen and Tajvidi, it has been proven 95   that the Extreme Value Theory may be used to represent extreme natural risks, such as floods, earthquakes, and wind-storm losses.

In the context of extreme value theory, it is possible to use the Weibull Law, a commonly used probability distribution, to simulate the exceedance probabilities of extreme events, such as geomagnetic storms. Coles and Katz et al. acknowledge the use of the Weibull Law in simulating extreme geomagnetic storms and associated concerns. The possible effects on various 100   businesses and infrastructure can be better understood using this method, especially in the context of insurance risk management. Extreme Value Theory has been shown to be helpful in predicting other extreme natural catastrophes such floods, earthquakes, and wind-storm losses in earlier research by Dahen et al., Rootzen and Tajvidi, and others. The use of the Weibull Law in conjunction with the Extreme Value Theory to simulate extreme events like geomagnetic storms has been acknowledged as a promising approach for precisely determining the likelihood of such events and their potential effects, 105   particularly in sectors like insurance that depend on accurate risk assessments.

## 2.5. Risk assessment in the insurance industry

Risk assessment, which involves calculating the risk and potential financial impact of numerous hazards, is a vital component of the insurance sector. The capacity to design suitable pricing strategies, control capital needs, and preserve financial stability is made possible by accurate risk assessment for insurers. As the implications of geomagnetic storms become evident, the 110   insurance industry seeks advanced risk assessment techniques to estimate potential damages and the financial implications of these events (Embrechts et al., 1997). Risk assessment makes use of a number of statistical models and methods, such as probabilistic modeling and Extreme Value Theory. These models and methods enable insurers to precisely estimate the likelihood of extreme events, such as geomagnetic storms, and their potential economic impact on various sectors of the





economy and infrastructure. Utilizing these techniques, particularly in the context of Extreme Value Theory and probabilistic modeling, provides insurance firms with a thorough framework for determining effective risk management strategies and analyzing the risk associated with natural disasters. Beyond geomagnetic storms, it has been demonstrated that the use of the Extreme Value Theory in risk assessment is helpful for modeling a variety of other extreme natural hazards, such as floods, earthquakes, and wind-storm losses. According to studies like Embrechts et al., the application of Extreme Value Theory in the insurance sector provides a useful framework for accurately assessing and managing risk, which is crucial for preserving financial stability and guaranteeing the long-term viability of the insurance sector. Beyond the forecasting and evaluation of geomagnetic storms, the application of extreme value theory to modeling severe natural disasters has significant implications for the insurance sector.

Additionally, by include extreme value statistics in business interruption insurance, insurers will be better able to predict how much money will be lost as a result of natural catastrophes. In general, the insurance sector has been able to analyze and manage risk related with natural disasters more effectively thanks to the adoption of modern statistical models and procedures. A thorough framework for precisely estimating the risk associated with extreme occurrences, such as natural disasters, has been made available by the application of probabilistic modeling and Extreme Value Theory in the insurance sector. This has improved the ability of insurers to calculate prospective monetary losses and create efficient risk management plans, thus enhancing the stability of the insurance sector as a whole.

## 2.6. Effects of Geomagnetic Storms on the Insurance Industry: A Risk Management Perspective

Geomagnetic storms present profound challenges for insurers due to their potential to cause widespread infrastructural damage, leading to significant economic repercussions. The insurance industry must grapple with claims from business interruptions, loss of revenue to repair costs, and liability suits from organizations impacted by outages (Schrijver et al., 2014; Eastwood et al., 2017). The broad geographic reach and unpredictability of these storms require innovative risk management strategies (Oughton et al., 2017; Kappenman, 2001).

## 3 Methodology

### 3.1. Data Sources and preprocessing

Dst index data from the World Data Center for Geomagnetism, spanning from January 1957 to 31.03.2023, were used in this study. The data were pre-processed to ensure quality and consistency and Dst values were converted to positive for ease of calculation. The onset of a geomagnetic storm with an intensity limited by the threshold occurs when the Dst signal reaches a certain negative threshold from above; the storm stops when the signal crosses the threshold once more. The thresholds



analyzed in this study range from 50 to 600 nT (nanotesla) in steps of 10 nT, but as will be explained in detail in the next section, the prediction models are used for storms lower than 150 nT and 300 nT.

## 3.2. Fitting the Generalized Extreme Value (GEV) distribution

A family of continuous probability distributions called the Generalized Extreme Value Distribution (GEV) can simulate the highest or lowest values in a dataset. Another probability distribution utilized in extreme value analysis is the Generalized Pareto Distribution (GPD), notably for modeling the tail of the distribution (exceedances over a threshold).

The Akaike information criterion (AIC) is a metric used to assess how well various statistical models fit the data. The model's complexity and likelihood given the data are both taken into account by the AIC. Better-fitting models are indicated by lower
AIC values.

Checking the extreme value model's assumptions and goodness of fit is referred to as investigating the model in EVA. This could include statistical analyses, diagnostic tests, and visual evaluation of the data and model fit.

The process of determining a statistical model's parameters based on observable data is known as fitting a model. The best model must be chosen, and its parameters must be optimized to reduce the discrepancy between the anticipated values and the
actual values. We used the maximum likelihood estimation method to fit a GEV distribution to the Dst data in order to model the extreme values of geomagnetic storm intensity.

Models utilized in EVT, such as the Block Maxima (BM) approach and the Peaks Over Threshold (POT) method, are referred to as Extreme Value Types. POT is used to model exceedances over a predetermined threshold, whereas BM is used to model the maximum (or minimum) values in fixed-size blocks of data.

Selecting the data points from the dataset that best represent extreme events is the process of extracting extreme values. To guarantee consistency, quality assurance and control procedures were used to the Dst index data that were collected from the World Data Center for Geomagnetism.

A critical component of understanding natural events, such as geomagnetic storms, is the investigation of extreme values. Researchers frequently use techniques like the Akaike information criterion and model fitting with the Generalized Extreme
Value distribution or Generalized Pareto distribution to accurately model the extreme values of geomagnetic storm intensity. In addition, validating assumptions and evaluating goodness of fit using numerous techniques, including diagnostic tests, statistical measures, and visual inspection of data, are all part of the investigation of the model in extreme value analysis.

To capture the tails of distributions, a careful investigation of extreme values is required since these exceptional occurrences can have a big impact on a variety of sectors, including finance, insurance, and telecommunications. In order to accurately
model and predict natural phenomena like geomagnetic storms, the study of extreme values is essential. To ensure accurate parameter estimation, researchers must use suitable statistical models like the Generalized Extreme Value distribution or the Generalized Pareto Distribution. An adequate model must be carefully chosen, and its parameters must be optimized, in order



to appropriately represent the extreme values of geomagnetic storm intensities. In order to assure consistency and accuracy, the process of extracting extreme values must be done carefully and with quality control procedures in place.

### 3.3. Parameter estimation using maximum likelihood method.

For optimal fit, the maximum likelihood method was used to estimate the parameters of the GEV distribution (Coles, 2001). The location, scale, and shape characteristics of the GEV distribution were all estimated using the maximum likelihood estimation method. A widely used method for estimating parameters in extreme value analysis is the maximum likelihood approach.

The location, scale, and shape parameters of the generalized extreme value distribution are all estimated using the rigorous and well-proven maximum likelihood estimation method. Due to its success in precisely capturing the behavior of extreme events, this method has been frequently employed in extreme value analysis. It entails maximizing the probability function of the observed data given a certain set of parameters. Additionally, researchers can choose the distribution that best fits a given dataset using methods like the Akaike information criterion and model fitting. For effective modeling of extreme values in a variety of industries, including the forecasting of geomagnetic storms, it is essential to choose the right model and estimate the parameters using the maximum likelihood technique.

### 3.4. Return level analysis and confidence intervals.

### 3.4.1 Estimating Return Periods

### 3.4.1.1 Return Period

Return period refers to the length of time (usually years) that reflects the likelihood that a particular value, such as wind speed, will be exceeded at least once per year. The probability of exceedance is referred to as this probability, and it is inversely related to return periods as 1/p, where p is the return period.

The return period analysis is a crucial component of extreme value analysis, and it aids in quantifying the likelihood of occurrence of occurrences that surpass a specific threshold.

Return periods are sometimes misunderstood in the professional world as "100-year event is an event which happens only once in 100 years," which can result in erroneous risk assessment. To view this more comprehensively, think about the time frame for risk assessment. Using the formula $P = 1 - (1-p)n$, where P is the probability of exceedance over n years, it can be determined that a 100-year event with a 1% likelihood of exceedance in any given year would have a probability of 39.5% to be surpassed at least once within 50 years.

To effectively calculate the danger posed by extreme occurrences in a variety of domains, including space weather, the estimation of return periods and accompanying confidence intervals is essential.



### 3.4.1.2 Probability of Exceedance

Extreme events extracted using BM or POT methods are assigned exceedance probabilities using the following formula Eq. (1):

$P = \frac{r-\alpha}{n+1-\alpha-\beta}$,                                       (1)

where:

·     r - rank of extreme value (1 to n).

·     n - number of extreme values.

·     Alpha($\alpha$) and Beta($\beta$) - empirical plotting position parameters

In this context P corresponds to a probability of exceedance of a value with rank r in any given time period with duration t/n where t is the total duration of series from which the extreme values were drawn and n is the number of extreme events. Extreme value analysis relies heavily on return level analysis, which gives an assessment of the likelihood that occurrences surpassing a specific threshold will occur on a regular basis. The estimation of return periods, which stand for the length of time during which a specific event is projected to occur based on historical data, is a useful function of return level analysis,

which is a key tool in estimating the risk associated with extreme events.

### 3.4.1.3 Return Period Calculation

Return periods are calculated from the exceedance probabilities using the following formula Eq. (2):

$R = \frac{1}{P}/\lambda$,                                                   (2)

where:

·     R - return period as multiple of return period size. (1 year)

·     P - $\frac{n}{t/return\_period\_size}$ exceedance probability calculated earlier.

·     $\lambda$ - rate of extreme events (average number of extreme events per return period size). Calculated as: $\lambda$ = or Peaks Over

Threshold, where n is number of extreme events and t is total duration of series from which the extreme values were drawn

Therefore, the resulting return period R is a real number that is a multiple of the return period size. As it provides an estimate of the time frame during which an event of a certain magnitude is anticipated to occur based on past data, the return period concept is a crucial tool in estimating the risk associated with extreme events.



## 4 Results

### 4.1. Fitted GEV distribution parameters.

Maximum likelihood estimate was used to get the fitted GEV distribution parameters, which served as the foundation for our probabilistic projections of geomagnetic storm episodes resembling the storm in March 1989.

First, we use the POT approach to remove severe occurrences with a minimum 24-hour gap between them from the hourly historical Dst index from the input time series. Our thresholds are 150 nT and 300 nT. In order to identify exceedance values that are roughly linear between thresholds, Mean Residual Life plots for threshold selection are created.

To determine where parameters are stabilizing around threshold levels and where following values have higher variance due to fewer exceedances, parameter stability tests utilizing the shape and scale parameters are undertaken.

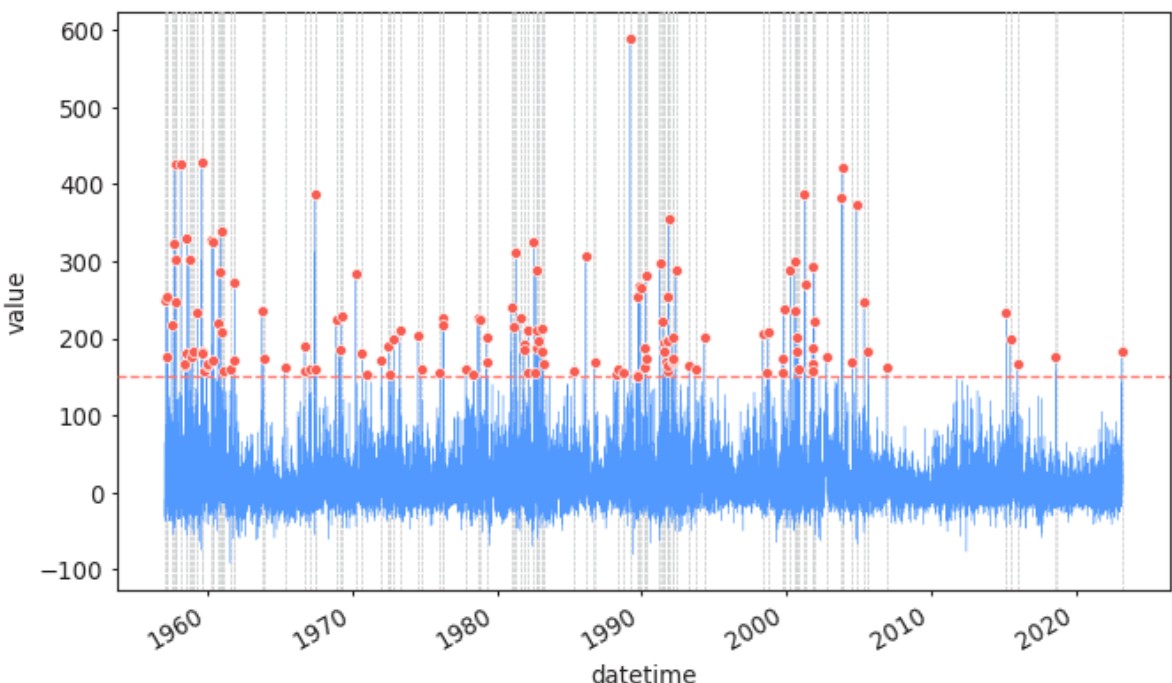

**Figure 1: Extracted Extreme Values Using the Peaks Over Threshold (POT) Method for 150 nT**

The events depicted in Figure 1 surpassing previously established benchmarks are highlighted when a threshold of 150 nT is utilized. The graph highlights the importance of extreme values within the specified threshold in the given context.

During our investigation into the 150 nT threshold, the visualization of the Mean Residual Life demonstrated a significant linear relationship in the exceedance values within the range of threshold values from 250 nT to 350 nT. The phenomenon of Parameter Stability was observed to exhibit a tendency to stabilize at approximately the threshold value of 250 nT. The





aforementioned stability was juxtaposed with a rise in variability in subsequent data points, which can be attributed to a decrease in the number of instances surpassing a certain threshold.

**Figure 2: Comprehensive Evaluation Including Shape, Scale, Parameter Stability, Return Value Stability, and AIC Optimization for 150 nT**





This visualization shown in Figure 2 effectively illustrates the complex relationship between these factors, which is crucial for making informed decisions on threshold selection.

The findings based on the analysis of historical data including shape, scale, Mean Residual Life and Parameter Stability show that the model exhibits robust stability for thresholds exceeding 300 nT, as suggested by Return Value Stability. The optimal

threshold range of 310 to 330 nT was determined by the Akaike Information Criterion (AIC). After a thorough evaluation, a consensus was reached on a threshold of 325 nT as the initial reference value. This decision was taken in order to determine the boundary of the next threshold analysis and to ensure the consistency of the model, considering the threshold at which the event examined in the model starts to become effective.

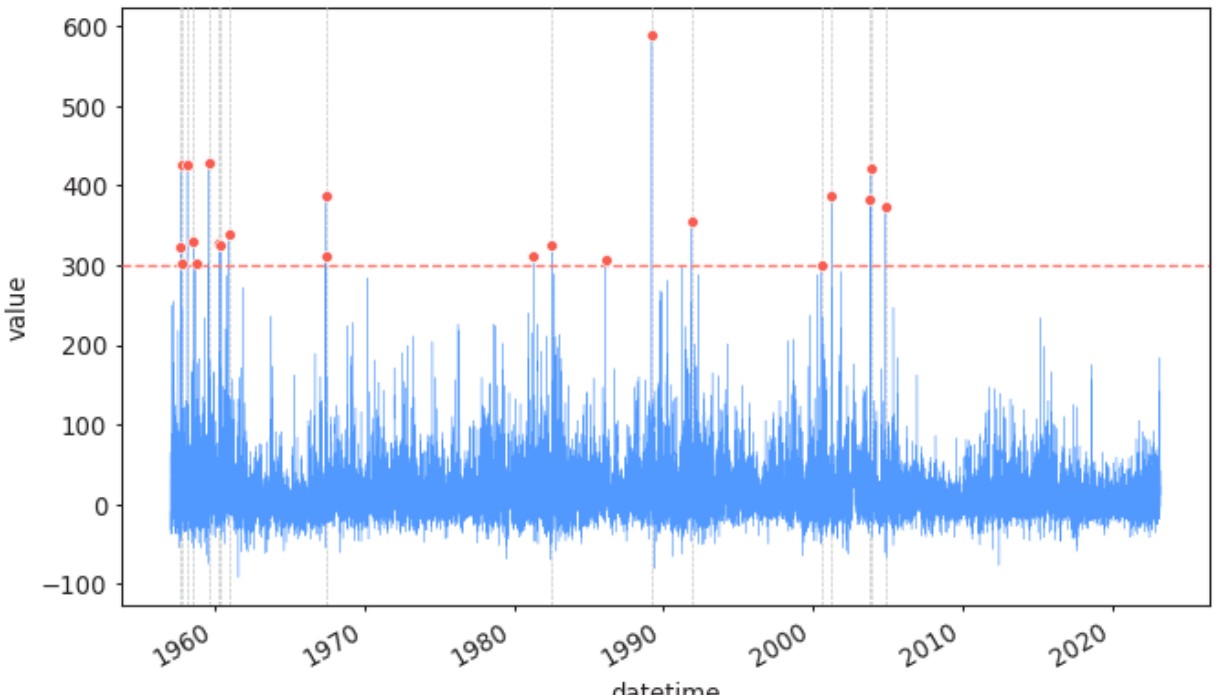

**Figure 3: Extracted Extreme Values Using the Peaks Over Threshold (POT) Method for 300 nT**

Depiction of events surpassing the historically established thresholds below 300 nT, highlighting the importance of exceptional occurrences shown in Figure 3.

The visualization of Mean Residual Life demonstrated a linear relationship in the exceedance values within the range of threshold values from 310 nT to 350 nT, with respect to the 300 nT threshold. The stability of the parameter exhibited

stabilization around the threshold of 300 nT, as evidenced by a subsequent increase in variance for subsequent values. This increase in variance can be attributed to a decrease in the number of exceedances.



**Figure 4: Comprehensive Evaluation Including Shape, Scale, Parameter Stability, Return Value Stability, and AIC Optimization for 300 nT**

Based on an analysis of the Shape, Scale, Mean Residual Life, and Parameter Stability metrics in our dataset, it is apparent that thresholds exceeding 330 nT exhibit a significant level of stability. The AIC directed our attention to a critical threshold range of 320 to 340 nT. After a thorough examination of the indicators shown in Figure 4, we arrived at a specific threshold value of 330 nT, thus placing our research on a solid second footing in terms of methodological rigor.




## 4.2. Estimated return periods

The estimated return periods were calculated using the POT method and the Weibull distribution, which allowed us to identify the most significant return periods for extreme geomagnetic storm events.

### 4.2.1 Estimating Return Periods (A) POT: Dst 150 nT)

| | Datetime | Value (Dst) | Exceedance Probability | Return Period |
|---|---|---|---|---|
| 0 | 1989-03-14 02:00:00 | 589 | 0.071429 | 71.3422 |
| 1 | 1959-07-15 20:00:00 | 429 | 0.142857 | 35.6711 |
| 2 | 1957-09-13 11:00:00 | 427 | 0.214286 | 23.7807 |
| 3 | 1958-02-11 12:00:00 | 426 | 0.285714 | 17.8355 |
| 4 | 2003-11-20 21:00:00 | 422 | 0.357143 | 14.2684 |

**Table 1: Estimating Return Periods (A) POT: Dst 150 nT)**

Table 1 presents the return periods obtained from the Peaks Above Threshold (POT) technique for Dst 150 nT. The most intense event occurred on 14 March 1989 and showed a Dst value of 589. This event exhibits an exceedance probability of 0.071429, corresponding to an estimated return period of approximately 71.34 years.

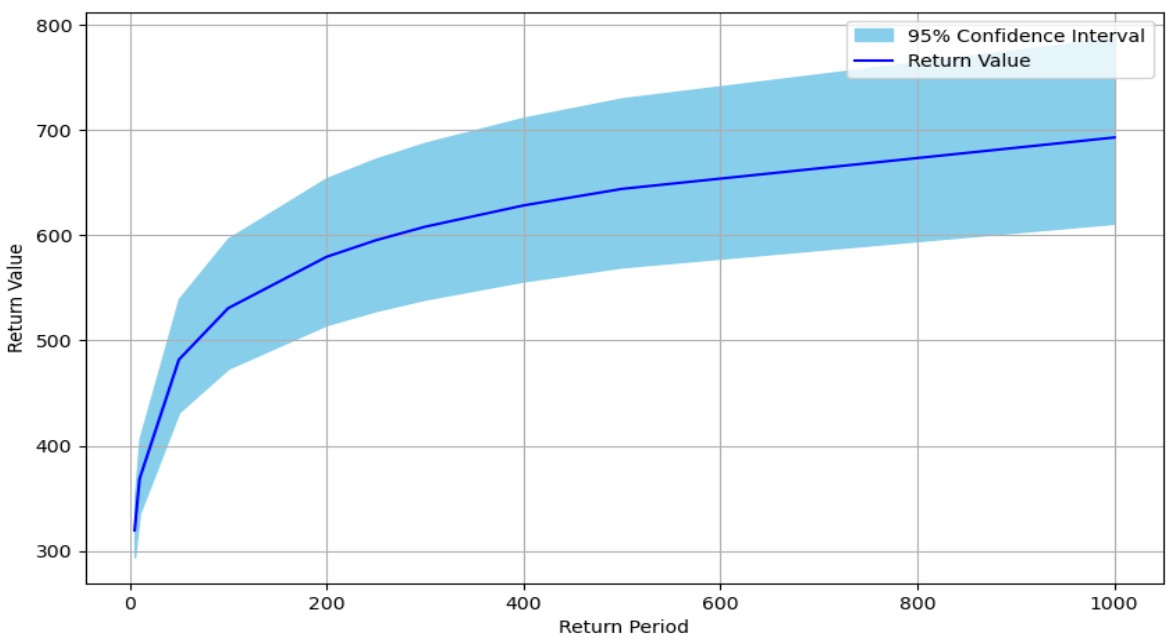

295

**Figure 5: Return Value against Return Period with 95% Confidence Interval for (A) POT: Dst 150 nT**





The purpose of Figure 5 is to visually represent the frequency at which geomagnetic disturbances of varying severity, as measured by the Dst index, are anticipated to occur over a specified time period. The threshold for Peaks Over Threshold (POT) is established at 150 nT. The 95% confidence interval, which is provided alongside each estimated return value, indicates a range within which the true return value is expected to lie with a 95% level of confidence.

### 4.2.2 Estimating Return Periods (B) POT: Dst 300 nT)

| | Datetime | Value (Dst) | Exceedance Probability | Return Period |
|----|---------------------|-------------|------------------------|---------------|
| 0 | 1989-03-14 02:00:00 | 589 | 0.083333 | 72.2687 |
| 1 | 1959-07-15 20:00:00 | 429 | 0.166667 | 36.1343 |
| 2 | 1957-09-13 11:00:00 | 427 | 0.25 | 24.0896 |
| 3 | 1958-02-11 12:00:00 | 426 | 0.333333 | 18.0672 |
| 4 | 2003-11-20 21:00:00 | 422 | 0.416667 | 14.4537 |

**Table 2: Estimating Return Periods (B) POT: Dst 300 nT)**

Table 2 presents the return periods associated with the Dst threshold of 300 nT. The observed data pattern exhibits a notable resemblance to the Dst threshold of 150 nT, albeit with certain significant variations. The most exceptional occurrence within this dataset also took place on March 14, 1989. However, it exhibits a slightly elevated exceedance probability of 0.083333, indicating a slightly longer return period of approximately 72.27 years.

It is imperative to acknowledge the differentiation in return periods and exceedance probabilities presented in Table 1 and 2. The aforementioned disparities highlight the significance of choosing a threshold that not only aligns with the dataset but also fulfills the requirements of the particular applications and desired level of sensitivity in the analysis.

In conclusion, the selection of thresholds has a substantial influence on the interpretation of both the frequency and severity of geomagnetic disturbances. According to the analysis conducted, it is evident that the implementation of a threshold of 150 nT leads to a greater occurrence of severe disturbances, albeit with reduced frequency. Conversely, the adoption of a 300 nT threshold implies a more permissive stance towards disturbances, classifying them as more frequent.





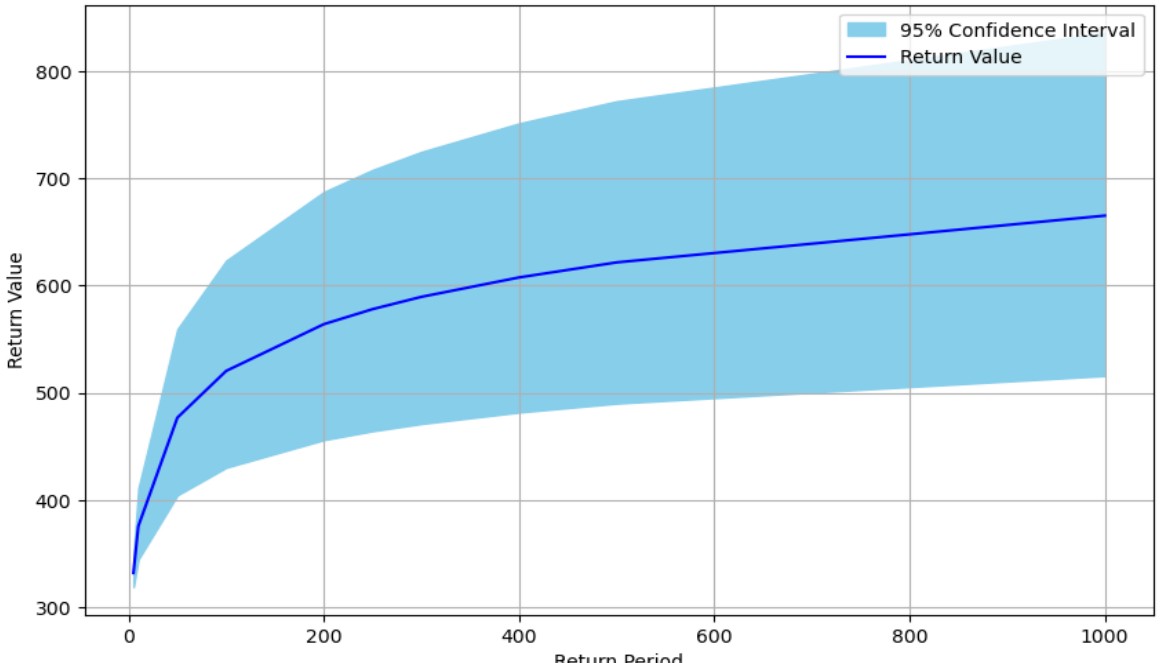

**Figure 6: Return Value against Return Period with 95% Confidence Interval for (B) POT: Dst 300 nT**

In a similar vein, this graphical representation illustrates the correlation between the return value and its corresponding return

period, albeit with a higher threshold of 300 nT. This implies that our analysis encompasses geomagnetic disturbances of a more severe nature. The utilization of a 95% confidence interval will facilitate comprehension of the level of uncertainty linked to these estimations of return values.

Through a comparative analysis of these two representations, valuable insights can be derived regarding the impact of the selected threshold for the Peaks Over Threshold (POT) method on the anticipated frequency and severity of geomagnetic

disturbances. Typically, a threshold of 150 nT would lead to more frequent disturbances of lesser intensity, whereas a threshold of 300 nT would result in less frequent but more severe events. Comprehending these dynamics is of utmost importance for the development and implementation of preparedness and mitigation strategies aimed at addressing geomagnetic disturbances.

**4.3. Probabilistic assessment of a March 1989-like geomagnetic storm**

Based on the fitted GEV distribution, EVT, the principles outlined by Weibull (1951), and POT technique, the likelihood of a

geomagnetic storm similar to March 1989 occurring over the next 70 years was calculated to be between 7.14% and 8.33% (with 95% confidence). Exposure determination associated with extreme weather events, such as geomagnetic storms, is based on estimating return periods using the probability and rate of exceedance of extreme events.



## 4.4. Sensitivity analysis and validation

By adjusting the threshold value and analyzing the influence on probability estimations, we conducted sensitivity analysis to
evaluate the robustness of our findings. To corroborate our conclusions, we also compared our results to those of previous
studies. In order to estimate the return periods of extreme geomagnetic storm events, the POT technique with the Weibull
distribution was employed in conjunction with historical data on extreme geomagnetic storm events.

Values of the Return Value Plots and Probability Density Plots of selecting the threshold values 325 and 330 are consistent
with each other and the frequencies move in the same direction from dense to sparse according to the POT method, which
shows that the thresholds selected in the study are consistent.

## 5 Discussion

### 5.1. Comparison with previous studies

Our findings support earlier research on geomagnetic storm probability, such as those by Riley (2012) and Thomson et al.
(2011), demonstrating once again the value of EVT in determining the likelihood of extreme geomagnetic events. This study
emphasizes the value of applying statistical techniques to determine the likelihood and danger of extreme weather phenomena
like geomagnetic storms. The POT approach and the Weibull distribution to calculate return periods were used in the study to
undertake a probabilistic analysis of a geomagnetic storm similar to the one in March 1989. Our findings enable more effective
application of updated risk assessment methods for geomagnetic storms in the context of the insurance industry.

### 5.2. Implications for risk assessment and management

The predicted likelihood of geomagnetic storms similar to those that occurred in March 1989 will have a big impact on the
insurance sector. In the face of potential losses brought on by extreme geomagnetic storms, accurate risk assessment and
management can assist insurers in creating suitable pricing strategies, allocating capital effectively, and maintaining financial
stability. Accurate risk assessment and management are also essential for companies that rely on these technologies since
geomagnetic storms can interfere with satellite and communication systems.
Geomagnetic storms and their potential impact require insurers to reconsider their risk management strategies. As these storms
present extensive infrastructural damage risks, leading to significant economic consequences, the insurance industry must
evolve its risk assessment methods and policy structures to factor in these unique challenges (Schrijver et al., 2014; Oughton
et al., 2017).



### 5.3. Prospects for Early Warning of Specific Geomagnetic Storms

**5.3.1. The Scientific Rationale for Early Warning Systems**

Geomagnetic storms are naturally occurring phenomena that are primarily induced by solar activities. Although significant progress has been made in comprehending these phenomena and their precursors, such as solar flares and coronal mass ejections, accurately forecasting their specific scale and consequences continues to pose a formidable task. Satellite-based monitoring of solar activity has facilitated the establishment of early warning systems. The aforementioned satellites possess 375 the capability to offer advanced warning prior to the arrival of storm effects on Earth, thereby granting critical infrastructure components a period of time to undertake precautionary measures.

### 5.3.2. Systems for Operational Forecasting

The current early warning systems primarily focus on specific critical components, including governmental institutions, nuclear power plants, military commands, and other essential assets. Nevertheless, a significant portion of industries, including 380 various sectors within the insurance field, exhibit a deficiency in implementing comprehensive strategies to mitigate the impact of geomagnetic storms. The existence of this discrepancy can be attributed to the inherent difficulties associated with predicting infrequent and highly intense occurrences, as well as the subsequent expenses associated with implementing comprehensive protective measures.

### 5.3.3. The Integration of Forecast-Based Action within the Insurance Industry

The insurance sector has historically exhibited a prudent approach in addressing unmodeled catastrophic incidents such as geomagnetic storms. Presently, these occurrences are regarded as unquantified uncertainties, with regards to both pricing and reserving. Consequently, the aforementioned storms continue to be predominantly overlooked in the risk management strategies employed within the sector.

Nevertheless, with the increasing advancements in modelling capabilities, the insurance industry has the potential to 390 significantly contribute to the promotion of mitigative measures. Insurers have the ability to encourage clients to implement direct physical, technological, or operational mitigation measures by incorporating the potential risks into their pricing and reserving mechanisms. Potential strategies that could be employed encompass modifying pricing structures, implementing exclusions, establishing sub-limits, or mandating preconditions for obtaining coverage.

The insurance sector possesses significant potential in terms of its capacity to enhance awareness and motivate clients to 395 comprehend and respond to these risks. As the advancement of modelling techniques for natural catastrophes progresses, insurers have the opportunity to integrate the resulting insights into their underwriting processes, thereby encouraging clients to prioritize mitigation measures. These incentives can be observed in both technological and operational spheres.



The insurance sector finds itself at a critical juncture, possessing the potential to exert a substantial impact on society's ability to effectively anticipate and alleviate the consequences of geomagnetic storms. By utilizing advancements in modelling and implementing forecast-based actions, the sector has the potential to take a leading role in not only protecting its own interests but also in fostering a more comprehensive societal resilience against these formidable natural events.

## 5.4. Limitations and future research

Although this study offers insightful information about the likelihood of extremely powerful geomagnetic storms, it has some drawbacks. To increase the precision and robustness of the estimations, more investigation might examine the effects of other threshold values and model presumptions, as well as include additional geomagnetic indicators and data sources.

Future study should also concentrate on creating more complex and thorough risk assessment models that take into account both the direct and indirect consequences of geomagnetic storms, such as on the electrical grid and communication systems. Accurate risk assessment and management are essential for a variety of industries, as shown by Thomson et al.'s study on the probabilistic assessment of extreme geomagnetic storms using statistical methods, and additional research in this area may result in more potent mitigation techniques.

## 6 Conclusion

In this study, we used extreme value theory to calculate the likelihood of geomagnetic storms of a similar intensity to the event in March 1989 and to analyze the effects of such storms on the insurance sector. According to our research, there is a 7.14% to 8.33% (with 95% confidence) chance that a geomagnetic storm of similar strength will occur over the next 70 years. These findings aid in our comprehension of the occurrence and severity of extreme geomagnetic storms and assist risk managers in the insurance sector. In the face of potential losses brought on by extreme geomagnetic storms, accurate risk assessment and management can assist insurers in creating suitable pricing strategies, allocating capital effectively, and maintaining financial stability. The study emphasizes the value of statistical techniques in determining the likelihood and risk of extreme weather occurrences, particularly geomagnetic storms.

## Competing interests

The contact author has declared that none of the authors has any competing interests



**Acknowledgments**

We would like to express our gratitude to the World Data Center for Geomagnetism, Kyoto, for providing the Dst index data used in this study. We also thank Dr. Johanna Scheller, Dr. Sebastian Pichler, Allianz Commercial Risk Management Team,
Arthur John Albertson, Michael Bruch and reviewers for their valuable feedback and suggestions, which helped to improve the quality of this work.

**Data Availability Statement**

Data from the World Data Center for Geomagnetism, Kyoto http://wdc.kugi.kyoto-u.ac.jp/ was used in the creation of this manuscript. Figures were made with Pyextremes version 2.3.1 (George Bocharov, 2023), available under the MIT license at
https://georgebv.github.io/pyextremes/. Part of the software (version 2.3.1) associated with this manuscript for the calculation is licensed under MIT and published on GitHub https://georgebv.github.io/pyextremes/ (George Bocharov, 2023).

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

License.