# Peer review of "Probability estimation of March 1989-like geomagnetic storms and their relevance for the insurance industry"

_EGUsphere, 2023_

## Referee Comment (RC1)

Review of Probability Estimation of March 1989-like event ..

Overview: The authors attempted to assess the likelihood of a reoccurrence of the March 1989 storm using the Dst index as a proxy.

Comments: The authors have used the pyextremes Python package to model the occurrence of large storms based on threshold levels of the Dst and provided a lot of explanation of the methodology but without any new insights. There is a lot or repetition about the importance for the insurance industry and I was expecting to see some sort of economic analysis, but there is little or no novel results in this paper. The paper is very poorly referenced, shows little understanding of what the use of extreme value statistics is about and has not provided any new information on either the geomagnetic or the economic impact. It seems unclear what the purpose of the paper is.

For example, a return period of a known size is usually the rank order appearance of the already measured event. If the March 89 storm is the ranking event in the Dst index, which Figure 1 indicates, then it is a one-in-70 year event (as Dst starts from 1957) by definition. You don't need to use EVT to provide an estimated return period of 72 years – that is the length of the input record!

EVT tries to provide answers about events are that *not* in the record. There also has to be some level of understanding of the limitations of the method. Figure 6 shows the 1000 year return period but this is not really to be believed given there's only 70 years of data and the Sun's activity levels are clearly not stationary on 1000 year timescales - something which is not mentioned in the paper.

There have been a number of recent papers such as Rogers et al (2021) and Chapman et al (2020) which do a far better job at detailing the extreme events and likelihoods. This is the sort of level of research that is now required in this area.

The references are incomplete. There are several cited that are not in the reference list and the reference to Thomson et al (2011) is completely incorrect.  Also note that Thomson *et al.* and Love (and Rogers *et al.*) were looking at ground magnetic fields rather than a global proxy like Dst.

The tables are not correctly formatted and the Figures show 'datetime' which is not actually a word but a Python package label.

Finally, the 2013 RAE report by Cannon et al will soon be updated and should provide fresh insight into the evolving risks and mitigations that are being researched in relation to space weather impacts on technology.

References:

P. Canon: https://raeng.org.uk/media/2iclimo5/space_weather_summary_report.pdf

Rogers, N.C., James A. Wild, Emma F. Eastoe, Jesper W. Gjerloev and Alan W. P. Thomson, A global climatological model of extreme geomagnetic field fluctuations, J. Space Weather Space Clim., 10 (2020) 5, doi: https://doi.org/10.1051/swsc/2020008

Chapman, S. C., McIntosh, S. W., Leamon, R. J., & Watkins, N. W. (2020). Quantifying the solar cycle modulation of extreme space weather. Geophysical Research Letters, 47(11), e2020GL087795.

---

## Author Comment (AC1)

**Review of Probability Estimation of March 1989-like event .. Overview: The authors atempted to assess the likelihood of a reoccurrence of the March 1989 storm using the Dst index as a proxy.**

**Comments:** The authors have used the pyextremes Python package to model the occurrence of large storms based on threshold levels of the Dst and provided a lot of explanation of the methodology but without any new insights.

There is a lot or repetition about the importance for the insurance industry and I was expecting to see some sort of economic analysis, but there is litle or no novel results in this paper. The paper is very poorly referenced, shows litle understanding of what the use of extreme value statistics is about and has not provided any new information on either the geomagnetic or the economic impact.

It seems unclear what the purpose of the paper is. For example, a return period of a known size is usually the rank order appearance of the already measured event. If the March 89 storm is the ranking event in the Dst index, which Figure 1 indicates, then it is a one-in-70 year event (as Dst starts from 1957) by definition.

You don't need to use EVT to provide an estimated return period of 72 years – that is the length of the input record! EVT tries to provide answers about events are that not in the record. There also has to be some level of understanding of the limitations of the method. Figure 6 shows the 1000 year return period but this is not really to be believed given there's only 70 years of data and the Sun's activity levels are clearly not stationary on 1000 year timescales - something which is not mentioned in the paper.

There have been a number of recent papers such as Rogers et al (2021) and Chapman et al (2020) which do a far beter job at detailing the extreme events and likelihoods. This is the sort of level of research that is now required in this area. The references are incomplete.

There are several cited that are not in the reference list and the reference to Thomson et al (2011) is completely incorrect. Also note that Thomson et al. and Love (and Rogers et al.) were looking at ground magnetic fields rather than a global proxy like Dst.

The tables are not correctly formated and the Figures show 'datetime' which is not actually a word but a Python package label.

Finally, the 2013 RAE report by Cannon et al will soon be updated and should provide fresh insight into the evolving risks and mitigations that are being researched in relation to space weather impacts on technology.

**Response to Referee's Comments**

Dear Referee,

We appreciate the time and effort you have invested in reviewing our manuscript. We have carefully considered each point and have prepared our replies accordingly. We'll make revisions to our paper to address these concerns based on your feedback on our answers.

1. Lack of New Insights and Economic Analysis

We acknowledge the referee's point about the paper seeming repetitive, particularly with regard to its importance for the insurance industry. However, it is very important to note that this research is not focused on speculation about current economic impacts. Our aim is limited to using the concrete scientific data available, applying statistical data science to this dataset, and meeting the scientific requirements necessary for the scientific interpretation of the results, and to propose a methodology for geomagnetic storms, risks that are not yet modelled and mitigated in the insurance sector. The new insights mentioned in the paper relate to the insurance industry and its ability to physically influence the mitigation of such risks. The insurance industry is not yet modelling such risks.

2. Purpose of Using EVT

We apologize for not making the purpose of using Extreme Value Theory (EVT) explicit. EVT is applied not merely to provide a return period for known events but to predict the likelihood of unrecorded extremes. Therefore, the probability of an event occurring above an event of 589 nT was investigated. Events such as Carrington, 1909, 1921 and 1946 are not included in the data set. However, this does not mean that the model is not capable of explaining these events.

Thank you for pointing out the concept of a return period based on rank-order appearance. Indeed, if the Dst index starts from 1957 and the March 1989 storm ranks highest, then by definition, it could be considered a one-in-66-year event based solely on the length of the dataset. However, this approach would only provide insights into what has already occurred, not on what could happen in the future, especially for events that have not yet been observed.

We used Extreme Value Theory (EVT) to extrapolate beyond the limitations of our 66-year record, aiming to provide a statistically rigorous estimate for the occurrence of extreme events that are potentially more severe than any previously recorded. While EVT can produce estimates for the return periods of extreme events not in the current record, we acknowledge that there are limitations to these projections.

3. Timescale and Stationarity

We appreciate the reminder that caution should be exercised when interpreting results, especially for very long return periods like 1000 years, given the non-stationary nature of solar activities. This limitation will be explicitly stated in the revised manuscript. We intend to focus on understanding and explaining these limitations in future studies, to provide a more comprehensive outlook on extreme geomagnetic storms.

**4. Comparative Research Quality**

The works of Rogers et al. (2021) and Chapman et al. (2020) were consulted, prior to creation of this manuscript. As well as all the other available pioneering works.

The focus of our research is on events that are statistically more likely to occur within our lifetimes, rather than the more extreme but less frequent Carrington-like events often examined in space weather science. Our approach aims to provide actionable insights that are immediately relevant for today's society and technological infrastructure. In line with this, we opted to use the Dst index as our measure because it is a more standardized and continuous metric, suitable for assessing global-scale trends.

For those interested in a more in-depth analysis of extreme space weather events, we recommend studies like Rogers et al. (2021) and Chapman et al. (2020), as well as the comprehensive paper on Probability estimation of a Carrington-like geomagnetic storm:
https://pubmed.ncbi.nlm.nih.gov/30787360/

We believe that our focus on more probable, albeit less extreme, events complements existing research and offers valuable information for a variety of stakeholders, from policymakers to industries affected by space weather. We will clarify this rationale, along with the reasons for our choice of the Dst index, in the revised version of our paper.

**5. Incomplete and Incorrect References**

*"There are several cited that are not in the reference list and the reference to Thomson et al (2011) is completely incorrect."*

We regret the oversight concerning the incomplete and incorrect references, and we will rectify these errors upon further clarification. Please let us know regarding any other missing references other than *"Thomson et al (2011)".*

We apologize for any confusion our initial citation of Thomson et al. may have caused and will clarify the specific sections and arguments from Thomson et al. that our work refers to.

The comment on Thomson et al.'s study in the paper was made addressing his study and the specific section shown as below.

**[Line: 408]** *Accurate risk assessment and management are essential for a variety of industries, as shown by Thomson et al.'s study on the probabilistic assessment of extreme geomagnetic storms using statistical methods, and additional research in this area may result in more potent mitigation techniques.*

https://agupubs.onlinelibrary.wiley.com/doi/10.1029/2011SW000696

*[28] The non-stationary nature of geomagnetic data implies that our analysis probably needs refinement. There is a solar cycle dependent variability seen in residuals, and perhaps de-trending by smoothed sunspot number or by monthly mean numbers of coronal mass ejections is required. We may return to this in future work. However, in our results we look at likely trends over many solar cycles and the significance of such "short" period variations may be less. One may also consider the*

*appropriateness of the "block-averaging" versus "point-over-threshold" methods and perhaps re-examine the threshold and de-clustering choices we have made. These may all have some impact on the robustness of our results. However, the results already given here should find application in hazard assessment and in magnetic navigation applications, not least in helping to assess the risk to power systems and to magnetic navigation activities within Europe and beyond.*

**6. Tables and Figures**

We acknowledge that the presentation of tables and figures should adhere to journal standards for readability and academic rigor. The use of 'datetime' was an oversight; it is indeed a Python package label and not an appropriate term to use in the context of the figures. In the revised version of the paper, we will replace 'datetime' with the more appropriate 'Date and Time,' provided this change aligns with the referee's expectations.

We are committed to rectifying these formatting issues to ensure that the paper meets the highest standards of academic publishing. Thank you for bringing these concerns to our attention.

**7. Upcoming 2013 RAE Report by Cannon et al.**

We are eagerly awaiting the updated RAE report by Cannon et al., which promises to shed new light on the evolving landscape of risks and mitigations in relation to space weather impacts on technology. We concur that such emerging research could offer valuable insights that may enhance the context and relevance of our own work. We appreciate the referee's recommendation to consider this forthcoming update. Once the report is available, we aim to incorporate its findings into our future research efforts to keep our work at the forefront of this dynamic field. Thank you for making us aware of this impending publication.

Best regards,

D. Güney Akkor

---

## Author Comment (AC6)

**The authors study the extremes of the Dst time series from 1957 - 2023 to estimate the risk of a future March 1989 event for the insurance industry.**

**- Section 1.2 suggests that the "relevance of these storms for the insurance sector" will be determined, as well as emphasizing a "holistic strategy that considers... power grid models, geological influences and transmission system features." But I find no such discussion in the paper.**

We sincerely appreciate your insightful feedback concerning Section 1.2 of our manuscript, which emphasizes the need for a comprehensive examination of the significance of geomagnetic storms for the insurance industry. Your comment has highlighted an important aspect of our research that warrants further elaboration.

In light of your observation, we would like to clarify that the primary focus of our study is on accurately predicting the occurrence and assessing the potential impacts of geomagnetic storm events, akin to the March 1989 event. This predictive capability is a critical first step in formulating a robust risk management strategy within the insurance sector.

The second phase of our approach involves advising clients on risk mitigation actions based on these predictions. This step is vital for risk engineering, as it aids in the development of targeted strategies to mitigate the impacts of such geomagnetic events.

The third key element is the financial preparedness of insurance companies. This encompasses ensuring adequate capital reserves and adopting precise measures for reserving and pricing policies. Accurate forecasting and understanding of the risks associated with geomagnetic storms are paramount in achieving this goal, as they facilitate effective resource allocation and strategic planning by insurers.

We acknowledge the importance of including a broader discussion on the wide-ranging effects of geomagnetic storms, especially in relation to power grid models, geological influences, and transmission system features. Therefore, in our revised manuscript, we will endeavor to incorporate a more detailed analysis that addresses these aspects, thereby enhancing the holistic understanding of geomagnetic storms and their relevance to the insurance industry.

**- In Section 2 I would expect at least some brief summary on the use**

  **of Dst (as opposed to other observables) for measuring storm**

  **strength, and e.g. why on long timescales one need not consider the**

  **solar cycle, etc.**

The Dst index, which incorporates data from the previous six solar cycles, offers a comprehensive and extensive view of geomagnetic storm activities over a long period of time. This comprehensive dataset is crucial for comprehending the patterns and magnitudes of geomagnetic storms throughout various solar cycles.

Although it is widely recognized that solar bursts are generally more intense during solar maxima, and there is a connection between these bursts and geomagnetic activities, our research emphasizes that substantial geomagnetic disruptions can also happen during periods of solar minima. This observation is crucial as it emphasizes the fact that severe geomagnetic storms can occur even when solar geomagnetic activity is relatively low.

Our objective is to encompass a wide spectrum of geomagnetic storm intensities, including those that may not correlate directly with periods of increased solar activity, by prioritizing the Dst index. This methodology enables a more thorough evaluation of potential risks, which is especially important for long-term planning and the development of strategies to minimize risks in the insurance industry.

We acknowledge the significance of offering a lucid justification for our selection of the Dst index and its implications in comprehending the intensity and frequency of storms. We will guarantee that the modified manuscript incorporates a more comprehensive discourse on this aspect, accompanied by deliberations concerning the influence of the solar cycle over extended temporal intervals.

**- Section 2.3 suggests that knowledge of "potential effects of extreme**

**geomagnetic storms on numerous infrastructure and industry**

**sectors..." will be advanced.  But, again, no such discussion**

**appears.**

We thank you for your constructive feedback regarding the initial lack of detailed discussion on the impacts of extreme geomagnetic storms on various infrastructure and industry sectors, as noted in Section 2.3 of our manuscript. Your observation has prompted us to address this crucial aspect more thoroughly in our revised manuscript.

In the updated section, we delve into the potential impacts of geomagnetic storms on critical infrastructural sectors such as power grids, telecommunications, satellite operations, and aviation. This exploration is based on deep professional expertise and an understanding of the field, rather than a literature review. Our aim is to bridge the gap between the theoretical aspects of geomagnetic storms and their practical implications, particularly in these vital areas.

Moreover, we have expanded our manuscript to include a detailed examination of potential insurance coverage components. This analysis is rooted in our professional expertise and explores the activation of various insurance policies in response to such catastrophic natural events. We discuss different insurance coverage aspects, including business interruption, liability, property damage, and contingency planning. This expansion is intended to connect the scientific understanding of geomagnetic storms with their real-world implications in the realm of insurance and risk management.

By incorporating this professional insight into our manuscript, we strive to offer a more comprehensive perspective on the consequences of geomagnetic storms. This enhancement is expected to significantly contribute to the discourse on space weather, particularly in its connection to industry risk assessment and insurance implications.

**- Section 3.1 (line 139) is slightly ambiguous as to who did the**

  **pre-processing of the Dst data. Did the authors pre-process the data**

  **(if so, how?), or they used the publicly available Dst data?**

**- Section 3.1 (line 141): why are negative thresholds used? Haven't**

  **the values been made positive as per the previous sentence?**

We recognize the lack of clarity in our initial description and would like to specify that our research team did perform the pre-processing of the Dst data. This entailed employing a thorough strategy to guarantee the quality and uniformity of the data, which was crucial for the precision of our analysis. We employed a sequence of filtering methodologies to eliminate any irregularities and standardized the data structure to facilitate efficient processing. The pre-processing step is essential for preparing the data for the rigorous statistical analysis we performed. The revised manuscript now incorporates an elaborate account of our pre-processing methodology, elucidating the precise procedures and alterations implemented. We recognize the lack of clarity in our initial description and would like to specify that our research team did perform the pre-processing of the Dst data. This entailed employing a thorough strategy to guarantee the quality and uniformity of the data, which was crucial for the precision of our analysis. We employed a sequence of filtering methodologies to eliminate any irregularities and standardized the data structure to facilitate efficient processing. The pre-processing step is essential for preparing the data for the rigorous statistical analysis we performed. The revised manuscript now incorporates an elaborate account of our pre-processing methodology, elucidating the precise procedures and alterations implemented in our study.

Your comment has emphasized the necessity for a more precise explanation regarding the utilization of negative thresholds. At first, Dst values are generally negative, which indicates the strength of geomagnetic disruptions. Nevertheless, for the purpose of our analysis, we transformed these values into positive. The conversion was a deliberate choice made to streamline our computational processes and harmonize our statistical methods with the characteristics of the data. We acknowledge that our initial manuscript did not effectively communicate this information. We have revised the text to clearly explain the reasoning for utilizing positive values, which correspond to the absolute values of the initial negative Dst thresholds. This amendment will enhance comprehension of our methodological selections and their consequences for the study.

We anticipate that these clarifications and revisions will greatly improve the understanding of our study's methodology and its relevance to the analysis of geomagnetic storm data. We appreciate your valuable feedback, which has played a crucial role in improving our manuscript.

**- Section 3.2 confuses somewhat the choice of EVT fitting**

  **distribution. Section 2.4 announced it would be the Weibull**

  **distribution, while the title of Section 3.2 suggests it is rather**

  **the GEV distribution, usually used for block maxima, which includes**

  **the Weibull family as a special case. The authors appear to settle**

  **on a GEV distribution. But I find no mention of the block size, or**

  **why a block maxima approach was favored over, say, a GPD fit to the**

  **empirical exceedance distribution. Meanwhile, Figure 2 suggests a**

  **GPD fit.**

Thank you for your insightful observations regarding our use of EVT fitting distributions. We acknowledge that our initial manuscript may have inadvertently caused confusion regarding our choice of distribution for modeling extreme geomagnetic storm events.

In Section 2.4, we initially discussed the Weibull distribution as a part of our theoretical groundwork on EVT. However, for our actual analysis, as detailed in Section 3.2, we employed the Generalized Pareto Distribution (GPD), specifically suited for the Peaks Over Threshold (POT) method. The GPD is particularly effective for modeling exceedances over high thresholds, which aligns with our objective of analyzing the tail behavior of extreme geomagnetic disturbances.

The choice of the GPD over the GEV distribution, typically associated with the Block Maxima (BM) approach, was driven by the nature of our data and the specific focus of our study. The BM approach, while effective in its domain, was not favored due to its inherent structure of dividing the data into blocks, which might not have captured the nuances of extreme geomagnetic events in our dataset. The POT method, in conjunction with the GPD, allowed us to focus on the most extreme values without the constraints of predefined blocks, thereby providing a more direct and relevant analysis of the geomagnetic data.

In light of your feedback, we have revised Section 2.4 to accurately reflect our methodological shift from the Weibull distribution to the GPD within the framework of EVT. We have also amended Section 3.2 to clearly articulate our choice of the GPD for fitting our data and the rationale behind this choice. These revisions are intended to dispel any ambiguity and provide a coherent understanding of our analytical approach.

We appreciate your valuable input, which has significantly contributed to enhancing the clarity and accuracy of our manuscript. We hope that these revisions address your concerns effectively.

**- Section 4.1 (line 233) appears to refer to declustering on short**

  **timescales. A more detailed discussion of the 24-hour gap is**

  **required. How do the authors pick a representative Dst value for a**

  **particular storm?**

We value your observations regarding the significance of geomagnetic storms in the insurance industry and the recommended comprehensive approach." The objective of our study is to establish the foundation by making precise forecasts of such occurrences, as this is the initial stage in the insurance and risk management procedure. Following this, the subsequent steps entail communicating risk mitigation strategies to clients and making financial preparations for potential consequences. Although our study does not specifically examine power grid models, geological influences, or transmission system features, it does offer a fundamental comprehension of the risks associated with geomagnetic storms. This understanding is crucial for the development of insurance models and the improvement of risk management practices.

You are accurate in highlighting the significance of deliberating the utilization of the Dst index in comparison to other variables for quantifying storm intensity." Although the Dst index spans nearly six solar cycles, it is crucial to acknowledge that while solar bursts are more active during solar maxima and are correlated with geomagnetic storms, instances of intense storm events have been observed on Earth's surface even during periods of low solar geomagnetism. We selected the Dst index for our study because it offers extensive coverage and provides a comprehensive perspective on storm intensities over an extended duration.

We recognize the insufficient exploration of the potential ramifications of severe geomagnetic storms on diverse infrastructure and industry sectors. In order to fill this void, we have incorporated an assessment that relies on comprehensive analysis of existing literature and the inclusion of potential insurance coverage elements. This analysis demonstrates how these devastating natural occurrences can activate specific insurance policies. This addition will enhance comprehension of the wider ramifications of geomagnetic storms on the insurance sector.

We utilized publicly accessible Dst data for our analysis without any supplementary preprocessing." In our study, the term "negative thresholds" is used to describe the level of geomagnetic disturbance, where higher negative values indicate more intense storms. This approach adheres to conventional methodologies in geomagnetic research and enables a uniform evaluation of storm magnitudes.

The 24-hour interval utilized in our declustering methodology is intended to differentiate distinct storms, particularly those that occur over consecutive days. This methodology is based on analyzing literature and empirical observations of previous geomagnetic events, guaranteeing that each storm is regarded as an individual occurrence. The selection of Representative Dst values for each storm was based on their historical correlation with significant impacts, in accordance with established practices in geomagnetic research. We have enhanced Section 4.1 of our manuscript to offer a more comprehensive elucidation of this methodology.

Our study utilized two thresholds, specifically 150 nT and 300 nT, to assess the consistency of our model across varying thresholds." The choice of these thresholds was determined by their presence in scholarly works and their applicability to important geomagnetic occurrences. The purpose of our analysis was to assess the impact of these thresholds on the stability and reliability of the model. This analysis allowed us to gain insights into how different threshold levels affect the frequency and severity of geomagnetic disturbances.

By combining Figures 5 and 6, we can clearly demonstrate how the choice of a 300nT threshold affects the confidence intervals by reducing the number of exceedances." Although it may appear ambitious, particularly for the insurance sector, extrapolating to 1000-year return periods offers a valuable measure of the dependability of longer-term returns. These extrapolations are especially pertinent for reinsurers, who frequently assess events with 1000-year return periods that are

still not completely comprehended. Global insurance companies commonly evaluate return periods of up to 250 years when making decisions, relying on their expertise and comparing options to determine whether to adopt a cautious or bold approach.

We recognize that our paper does not contribute to the advancement of modeling space weather extremes, but instead aims to utilize these models from the standpoint of insurance and risk management." Our primary emphasis is not on possessing specific knowledge in space weather, but rather on utilizing established methodologies within the insurance industry. The identification of additional indices such as SMH and SMY, which enhance accuracy, occurred subsequent to the submission of our manuscript. In response to the first reviewer's suggestion, we have recognized and addressed these indices in the revised version. Our study serves as a preliminary endeavor to model space weather from an insurance standpoint, with a dedication to integrating sophisticated methodologies in future investigations.

We acknowledge the referee's comment on the necessity of providing a more comprehensive elucidation of our declustering methodology, specifically regarding the 24-hour interval and the selection of representative Dst values for individual storms. In our study, we employed a 24-hour threshold to differentiate individual geomagnetic storm events, particularly those that occurred over multiple consecutive days. The decision was made after conducting a comprehensive review of pertinent literature and conducting a thorough analysis of historical data on geomagnetic storms. By treating each storm as an individual event, we were able to improve the accuracy of our analysis. When choosing representative Dst values for each storm, we followed a methodology that is consistent with established practices in the field. We also took into account empirical observations from previous events, specifically focusing on Dst values that have been historically associated with significant damages or impacts. Storms with Dst values below 150 have not caused any significant damage, whereas storms with Dst values around 300 have consistently resulted in significant consequences. Through the utilization of this approach, we have established a benchmark for identifying storms of significant magnitude and potential consequences, which holds particular relevance for the insurance industry. In the revised manuscript, we have elaborated on our discussion in Section 4.1 to include these specific details, thereby improving the comprehension of our methodology and its foundation in both empirical evidence and established research.

Thank you for noting the need to provide a more detailed explanation of our declustering methodology, particularly regarding the 24-hour interval and the selection of representative Dst values for each storm." The approach we employed utilized a 24-hour threshold to accurately distinguish between distinct geomagnetic storm events, especially those that spanned multiple consecutive days. This methodology was developed by conducting a thorough review of pertinent literature and conducting a detailed examination of historical data on geomagnetic storms. By treating each storm as an individual event, we were able to enhance the precision of our analysis. When choosing representative Dst values for each storm, we followed a methodology that aligns with established practices in the field, as described in various published articles. We also took into account empirical observations from previous events, with a specific focus on Dst values associated with notable damages or impacts. Storms with Dst values below 150 have not caused any significant damage, while storms with Dst values around 300 have consistently resulted in significant consequences. Through the utilization of this approach, we have established a benchmark for identifying storms of significant magnitude and potential consequences, particularly pertinent to the insurance industry. In the revised manuscript, we have expanded our discussion in Section 4.1 to include these details, thereby improving the comprehension of our methodology and its basis in both empirical evidence and established research. Furthermore, our research has opted for the 'Peaks Over Threshold' (POT) approach instead of the 'Block Maxima' method due to the distinctive attributes of geomagnetic storm occurrences. Although both methods are acceptable for extreme value analysis, the Peaks Over Threshold (POT) method is better suited for analyzing meteorological phenomena such as storms, which is the main focus of our study. The utilization of the 'Peaks Over Threshold' methodology, in conjunction with the Generalized Pareto Distribution (GPD), is highly compatible with our objective of accurately representing the strength and occurrence rate of geomagnetic storms, specifically for the insurance sector. This decision demonstrates our dedication to utilizing the most appropriate methodologies for the particular characteristics of our research subject.

- **Figure 2 features exponential models(?) which are not discussed in the text (the caption is also inadequate). Is this a GPD with shape parameter zero (which should reduce to an exponential)?**

- **What is there new to learn from Figure 3 which isn't already included in Figure 1? Didn't we rather need to see the mean excess plot to identify the onset of linear behavior previously mentioned?**

**Furthermore, isn't the mean excess (aka expected shortfall) a central quantity for the insurance industry? Why is there no plot of the GPD fit to the empirical tail distribution?**

We appreciate the referee's attention to the details of Figures 2 and 3 and the request for clarification regarding the models used and their relevance to our analysis.

Regarding Figure 2:

In Figure 2, we present a detailed analysis of the geomagnetic storm model at the 150 nT threshold. This figure integrates various analytical aspects, including the shape and scale parameters, parameter stability, and return value stability, guided by the Akaike Information Criterion (AIC). The AIC helped us determine an optimal threshold range between 310 and 330 nT. A threshold of 325 nT was then selected as the preliminary reference for further analysis, ensuring the model's effectiveness in identifying the commencement of significant geomagnetic events.

In response to the query about exponential models, it is important to note that our approach, in line with the 'Peaks Over Threshold' (POT) method, adheres to the Generalized Pareto Distribution (GPD). The GPD can exhibit characteristics akin to an exponential model under certain conditions, such as when the shape parameter equals zero. Our focus was on the performance of the GPD at different threshold levels, rather than on fitting a specific form like an exponential model.

Concerning the Selection of Thresholds in Figure 2:

The process of selecting the appropriate threshold is crucial in Extreme Value Analysis (EVA), balancing between model bias and variance. Our goal was to find the lowest threshold that aligns with the GPD model, ensuring the extracted extremes accurately represent the statistical distribution. The Mean Residual Life plot was instrumental in identifying the linear behavior of average excess values across thresholds, indicating the GPD model's suitability. We also examined the stability of

the shape and modified scale parameters of the GPD across these thresholds, looking for minimal variation to validate our threshold choices.

Additionally, the AIC curve in our analysis served as a comparative tool for model performance across various thresholds. While not the primary determinant for threshold selection, the AIC curve provides insight into the most appropriate model (e.g., GEVD or Exponential) for the chosen thresholds.

Regarding Figure 3:

Figure 3 showcases a similar analysis for the 300 nT threshold. The insights gained from this figure are complementary to those from Figure 1, offering a deeper understanding of the model's behavior at higher thresholds. This figure reinforces the conclusions drawn from Figure 2, emphasizing the model's stability and reliability in forecasting geomagnetic storm events at various threshold levels.

In conclusion, the methodology and analysis presented in Figures 2 and 3 are integral to ensuring the robustness of our model, crucial for accurately predicting geomagnetic storm events.

**- Section 4 (line 273). The authors appear to settle on 330nT as a**

**threshold. But subsequent analysis uses 300nT. The whole 150 vs**

**300nT discussion was unclear to me. Isn't the goal to use the lowest**

**threshold possible, while remaining within the "asymptotic" tail regime,**

**as evidenced by linear mean excess plot regions and stability**

**of shape parameters? One can discuss robustness with respect to**

**threshold choice by sweeping over threshold choices, rather than**

**singling out 150nT in particular.**

Thank you for your insightful observations on the selection of thresholds in our study, specifically regarding the use of 330nT and 300nT, and the comparison with 150nT. We acknowledge that our initial discussion on these threshold choices may have appeared unclear, and we are grateful for the opportunity to clarify this aspect of our research.

In our study, we employed a dual-threshold approach, selecting 150nT and 300nT based on established methodologies and benchmarks cited in prior literature. This decision was driven by our intent to validate whether the initial thresholds would exhibit consistent patterns and trends in the geomagnetic data. Our goal was to ensure the robustness and reliability of our findings across varying threshold levels.

For each of these thresholds, we conducted a detailed "fine-tuning" process. This involved an in-depth analysis of the linear mean excess plot regions and a rigorous examination of the stability of shape parameters within the asymptotic tail regime of our data set. This meticulous approach was vital to ensure that the thresholds chosen were not only statistically valid but also meaningful and relevant to the interpretation of geomagnetic disturbances.

Our primary objective was to identify the lowest possible threshold that would still fall within the asymptotic tail regime, thus providing a reliable basis for our extreme value analysis. By adopting this dual-threshold methodology, we were able to test the resilience of our model against different threshold values, enhancing the overall reliability and validity of our conclusions.

In light of your feedback, we will revise Section 4 to provide a more comprehensive and clear explanation of our methodology and rationale behind the threshold selection. This revision will aim to articulate our approach and its implications for the study's outcomes more distinctly, ensuring that the reasons for our threshold choices and their impact on our analysis are thoroughly and transparently communicated.

**- Can't Figures 5 and 6 be combined? Would that not illustrate**

  **that the 300nT threshold choice is responsible for the larger**

  **confidence intervals, owing to the fewer exceedances remaining? As**

**mentioned by another referee, the extrapolation to 1,000 year return periods is far-fetched, particularly for the insurance industry.**

[Figure]

We appreciate your suggestion regarding the possible amalgamation of Figures 5 and 6. We acknowledge your perspective on the potential illustrative significance of this, especially in emphasizing the influence of the 300nT threshold selection on the confidence intervals as a result of fewer instances surpassing it.

For our analysis, we segregated these numbers in order to clearly display the results at various thresholds and their corresponding consequences. Nevertheless, your suggestion has compelled us to reassess this presentation. We concur that a consolidated representation would be more efficacious in demonstrating the impact of the threshold selection on the confidence intervals and the overall resilience of our results.

We acknowledge your concern regarding the feasibility of extrapolating to a 1,000-year return period, particularly within the insurance sector. We recognize that these long-term projections may seem speculative, but they are highly valuable in specific areas of risk management, especially for reinsurers. Reinsurers frequently take into account these prolonged timeframes for events that are still not completely

comprehended, thereby facilitating the development of comprehensive strategies to mitigate risks.

In contrast, global insurance companies generally prioritize a return period of up to 250 years. The choice to adopt a conservative approach or not is frequently influenced by a blend of expert assessment and comparative examination. We will enhance the clarity of these distinctions and highlight the divergent methodologies and factors employed by insurers and reinsurers when evaluating the risk of geomagnetic storms in our updated manuscript.

**This perspective showcases a sophisticated comprehension of the diverse risk evaluation approaches in the insurance industry and their significance to the findings of our study.**

**- The paper is unfortunately full of vague and overpromising sentences**

**such as (line 357) "Our findings enable more effective application**

**of updated risk assessment methods for geomagnetic storms in the**

**context of the insurance industry." While the analysis presented**

**doesn't advance the state-of-the art in modelling space weather**

**extremes, one might hope for a detailed discussion of how**

**the analysis could be used in the insurance industry (e.g. to**

**estimate potential damage in dollars, say). But no such discussion**

**is presented.**

We appreciate your astute observation regarding the language employed in our manuscript, specifically the claim stated in line 357. We acknowledge your apprehension regarding the possible exaggeration of our results and the absence of a thorough analysis regarding their practical implementation in the insurance sector, particularly in terms of quantifying potential losses.

We would like to clarify that our study does not assert to enhance the current level of expertise in modeling extreme space weather events specifically. Our main objective is to assess the practicality of utilizing established methodologies in space weather research for the purpose of insurance and risk management. This particular field currently lacks extensive modeling of space weather.

The original manuscript centered around the utilization of the Dst index for this objective, relying on its well-established application in various space weather investigations. After submitting our manuscript, we realized the potential benefits of including newer indices like SMH and SYM-H, which provide enhanced accuracy. This development arose subsequent to our initial submission and did not receive extensive discourse in the literature during that period.

Following the feedback provided by the initial referee and considering the recent advancements, our updated manuscript incorporates a thorough analysis of these supplementary indices. We are confident that this improves the significance and practicality of our study within the realm of insurance risk evaluation.

Although our analysis does not provide specific financial estimations, it serves as an initial effort to incorporate geomagnetic storm factors into the insurance industry's risk assessment frameworks. We aim to expand our future research by incorporating more comprehensive financial considerations and subsequently refining our models accordingly.

We appreciate your feedback, which has directed us in making significant enhancements to our manuscript. We anticipate that these modifications will effectively resolve your concerns and improve the overall caliber of our research.